# A Self-Healable and Recyclable Zwitterionic Polyurethane Based on Dynamic Ionic Interactions

**DOI:** 10.3390/polym15051270

**Published:** 2023-03-02

**Authors:** Haiyan Mao, Qun Zhang, Ling Lin, Xuemei He, Lili Wang

**Affiliations:** School of Textile & Clothing, Yancheng Institute of Technology, Yancheng 224051, China

**Keywords:** zwitterionic polyurethane, elasticity, self-healing, reprocessing

## Abstract

Polyurethanes with self-healing and reprocessing capabilities are promising in eco-friendly applications. Here, a self-healable and recyclable zwitterionic polyurethane (ZPU) was developed by introducing ionic bonds between protonated ammonium groups and sulfonic acid moieties. The structure of the synthesized ZPU was characterized by FTIR and XPS. The thermal, mechanical, self-healing and recyclable properties of ZPU were also investigated in detail. Compared with cationic polyurethane (CPU), ZPU shows similar thermal stability. The physical cross-linking network formed between zwitterion groups can dissipate strain energy as a weak dynamic bond, endowing ZPU with outstanding mechanical and elastic recovery properties, including the high tensile strength of 7.38 MPa, high elongation at a break of 980%, and fast elastic recovery ability. Additionally, ZPU exhibits a healing efficiency of over 93% at 50 °C for 1.5 h as a result of the dynamic reconstruction of reversible ionic bonds. Furthermore, ZPU can be well reprocessed by solution casting and hot-pressing with a recovery efficiency above 88%. The excellent mechanical properties, fast repairing capability, and good recyclability not only enable polyurethane with a promising application in protective coatings for textiles and paints but also make it a superior candidate as stretchable substrates for wearable electronic devices and strain sensors.

## 1. Introduction

Polymeric materials are extensively employed in almost every area of modern life due to their low density, superior processability, excellent flexibility, and many other features. However, conventional polymers are easily damaged by external factors such as mechanical abrasion, thermal decomposition, radiation damage, etc., which influence their mechanical strength and reduce their service life and safety [1]. Meanwhile, a lot of waste produced from these unrecyclable polymers may result in serious environmental pollution due to their irreversible covalent bonds [2]. Thus, developing self-healing polymers is an efficient path to address these problems. Similar to the self-healing phenomenon of skin tissue, self-healing materials are able to repair physical damage and/or functions via the bonding and deboning of dynamic bonds in response to external stimuli, thereby significantly prolonging service life, improving reliability and endowing recyclability [3,4,5]. The dynamic bonds include reversible noncovalent interactions (such as hydrogen bonds, ionic bonds, mental-ion binding interactions, etc.) and reversible covalent bonds (such as disulfide bonds, imine bonds, boronic ester/boroxine bonds, Diels-Alder reaction, etc.). Among them, reversible noncovalent bonds are beneficial for designing the material with self-healing abilities under mild conditions, which is more energy-efficient and user-friendly. These self-healing materials can be widely employed in electronic skin, wearable electronics, sensors, protective coatings, and many other fields.

Polyurethanes are one of the most attractive polymeric materials due to their variable molecular structures, tailorable physicochemical properties, and outstanding processability [6,7]. They have been widely applied in textiles, coatings, adhesives, elastomers, foams, wearable devices, flexible electronics, and other areas [8,9]. According to charge property, polyurethanes are generally divided into nonionic, cationic, anionic, and zwitterionic types. Among them, zwitterionic polyurethanes are a kind of neutral polyelectrolyte with equal cationic and anionic groups within the same segment in polymer chains [10,11]. In general, the cationic group is quaternary ammonium, and the anionic moiety is sulphonate, carboxylate, phosphate, and so on. Zwitterions can form a hydration layer as a physical and energetic barrier on the material surface [12,13]. This structural feature endows the zwitterionic polyurethanes with many unique properties, such as super-hydrophilicity, good biocompatibility, and anti-fouling performance, and thus have been widely used in antifouling coatings, surfactants, membrane separation materials, biosensors, and biological medicine [14,15,16]. For example, Nikam et al., prepared a zwitterion-derivatized polyurethane with an allyl ether-functionalized PU and a zwitterionic thiol via a radically induced thiol-ene reaction [17]. This zwitterion-PU presented a remarkable reduction in bacterial adherence and excellent biocompatibility. Coneski et al., synthesized zwitterionic polyurethane hydrogels derived from carboxybetaine-functionalized diols, which showed tunable water uptake and low bacterial adhesion [18]. Huang et al., also designed a novel zwitterionic polyurethane with sulfobetaine zwitterionic moieties, which were used to coat acrylic polyurethane [19]. Due to the zwitterionic moieties, this coating showed excellent antibacterial properties and abrasive resistance. Liang et al., fabricated an antifogging coating by copolymerizing sulfobetaine methacrylate, 2-hydroxyethyl methacrylate, and sulfobetaine-modified silica nanoparticles [20]. This coating shows outstanding antifogging properties due to the strong hydration ability of the zwitterionic moieties.

Apart from the above widely studied anti-fouling and antifogging properties, the self-healing capability of zwitterionic materials has also attracted increased attention. Zwitterion can be reformed through intra- or intermolecular reversible ionic interactions to achieve molecular recombination under mild repairing conditions [21]. As one of the dynamic non-covalent bonds, ionic interactions possess a higher bond energy than hydrogen bonds and can further form ionic multiplets or clusters with a higher association energy than a single ionic bond [2]. For example, Bai et al., introduced a betaine monomer to prepare zwitterionic hydrogel through the double bond addition reaction, which can repair itself under physiological conditions without providing additional stimuli. Importantly, the self-healing behavior of the zwitterionic hydrogel has no time limit, and the healing efficiency can reach 90% [22]. Sun et al., prepared a physically crosslinked zwitterion nanocomposite hydrogel with carbon nanotubes and a poly (sulfobetaine methacrylate) network [23]. The nanocomposite hydrogel showed fast self-healing abilities with a healing efficiency of 82% after 3 min at room temperature. Chen et al., prepared a zwitterionic polyurethane with hexamethylene diisocyanate, N-methyldiethanolamine, and propane sultone, which not only showed excellent multiple shape memory but also could repair the fractured samples under humid conditions for a few minutes and then at 50 °C for 2 h [24]. Wang et al., also developed a zwitterionic polyurethane, which was used for a triboelectric nanogenerator; this polyurethane exhibited a tensile strength of 0.65 MPa, an elongation of 190% and a healing efficiency of about 89% [25]. However, the strength, stretchability, and the self-healing properties of these polymer materials usually cannot be balanced, which limits their widespread application. Therefore, it is still a great challenge to design self-healing polyurethane materials with both outstanding mechanical properties along with high healing efficiency under mild conditions.

In this paper, we reported a self-healable and recyclable zwitterionic polyurethane based on dynamic ionic bonds, which is easily prepared via a ring-opening reaction of propane sultone and tertiary amino groups. The chemical structure, thermal stability, mechanical, elastic recovery, self-healing, and recyclable properties of the prepared zwitterionic polyurethane are discussed in detail. This work provides a new strategy for developing zwitterionic polymers with excellent elastic, self-healing, and recyclable capabilities.

## 2. Materials and Methods

### 2.1. Materials

Isophorone diisocyanate (IPDI, 99%) was bought from Aladdin (Shanghai) Co., Ltd. *N*, *N*-bis (2-hydroxyethyl) methylamine (MDEA, 99%), 1,3-propanesultone (PST, 98%) and anhydrous tetrahydrofuran (THF, 99.5%) were all available from Energy Chemical. Polycarbonate diol (PCDL, Mn = 1000 g/mol) was purchased from Shandong Baiyi Chemical Co., Ltd. (Qingdao, China). Dibutyltin dilaurate (DBTDL) and glacial acetic acid (HAc) were supplied from Sinopharm Chemical Reagent Co., Ltd. (Shanghai, China). Additionally, PCDL were vacuum-dried at 100 °C for about 12 h prior to use. All other reagents were used without further purification.

### 2.2. Preparation of ZPU and CPU

The detailed synthetic route of zwitterionic polyurethane (ZPU) is shown in Figure 1. ZPU was prepared based on step-growth addition polymerization through two steps. Firstly, IPDI (8.89 g, 40 mmol), PCDL (20 g, 20 mmol), and the DBTDL catalyst (0.5 wt.%) were dissolved in anhydrous tetrahydrofuran and then fed into a three-necked flask equipped with a mechanical stirrer and a reflux condenser. After the mixture was reacted at 70 °C for 2 h, MDEA (2.4 g, 20 mmol) was dissolved in anhydrous tetrahydrofuran and then fed to the prepolymer solution with a further chain extending for 2 h. Anhydrous tetrahydrofuran (about 30 mL) was also added during the pre-polymerization and chain extension reactions to reduce the viscosity of the reaction system and ensure fast reactions. Next, PST (2.44 g, 20 mmol) was added and reacted at 70 °C for another 3 h to obtain viscous ZPU. Finally, the ZPU film was prepared by the solvent casting method on a Teflon mold, followed by drying at room temperature for 48 h and then at 60 °C for another 24 h to remove the THF solvent.

The controlled cationic polyurethane (CPU) sample was also synthesized using HAc to neutralize MDEA instead of PST. The proportion of other components, synthesis procedure, and conditions was the same as those of ZPU. The viscous CPU was then poured into a Teflon mold, and the THF solvent evaporated at room temperature for 48 h and then at 60 °C for another 24 h.

### 2.3. Characterization

Fourier transform infrared (FTIR) spectra were performed on an IR spectrometer (NEXUF-670) equipped with attenuated total reflectance (ATR). The measurements were carried out within the wavenumber range of 500–4000 cm^−1^ and a resolution of 4 cm^−1^ at room temperature.

X-ray photoelectron spectroscopy (XPS, PHI Quantum 2000) spectra were conducted using a monochromatic (Al Kα) X-ray beam. Survey scans, high-resolution C 1s, N 1s, and S 2p spectra were all recorded, and the data of high-resolution spectra were analyzed by XPSPEAK software.

The thermogravimetric analysis (TGA) of ZPU and CPU was recorded using a TGA instrument (STA449C) at 25–600 °C and a heating rate of 10 °C/min under a nitrogen atmosphere with a flow rate of 20 mL/min.

Tensile tests of all samples were performed on a CMT4304 universal tensile machine (MTS, China). A film (~30 mm length × 40 mm width × 0.5 mm thickness) was stretched at the rate of 50 mm/min under ambient conditions. Three measurements were carried out for each sample. Cycle tensile tests were performed at a loading/unloading speed of 100 mm/min at room temperature. Young’s modulus was calculated as the slope of the initial tangent line from the 10% strain of the stress–strain curves.

To evaluate the full-cut self-healing property, the ZPU film was cut into two pieces from the middle, and then the two pieces were aligned and connected together. Deionized water was added dropwise to the connected part of the film, which was then placed at 50 °C for different durations. The healed film was observed by a digital microscope and quantitatively evaluated the healing process through a tensile test. To quantitatively evaluate the healing effect, the healing efficiency (HE) was introduced and calculated based on the recovery of breaking strength or elongation at break, which is expressed as the following equation:HE = *T*_healed_/*T*_origianl_ × 100%(1)
where *T* represents the breaking strength or elongation at break.

In reprocessing experiments by solution casting, the ZPU film was cut into small pieces and then dissolved in THF at room temperature for 12 h, followed by pouring into a Teflon mold at room temperature for 48 h and then at 60 °C for another 24 h to remove the THF solvent. As for reprocessing by hot-pressing, small pieces of the ZPU film were put in a mold, followed by heating and compressing at 80 °C, 8 MPa for 5 min. The recovery efficiency was evaluated through tensile tests.

## 3. Results and Discussion

### 3.1. Fabrication and Structure Characterization

The zwitterionic polyurethane (ZPU) consists of hard segments and soft segments featuring a dynamic ionic interaction. Typically, isophorone diisocyanate (IPDI) and N, N-bis (2-hydroxyethyl) methylamine (MDEA) were used as the hard segments. Polycarbonate diol (PCDL) with a moderate molecular weight was employed as the soft segment, which ensures the molecular chain mobility and mechanical strength of the polyurethane. Finally, 1,3-propanesultone (PST) was incorporated to react with tertiary amino groups in MDEA via ring-opening reactions to generate the sulfobetaine zwitterionic structure. The zwitterions in the polyurethane chains form an ionically crosslinked network through electrostatic interactions to enhance the mechanical properties alongside the self-healable and recyclable capability of the polyurethane material.

The chemical structure of the synthesized ZPU was characterized by the Fourier transform infrared spectroscopy (FTIR) at first, as presented in Figure 2. For both ZPU and CPU, the peaks at 3357 cm^−1^, 1741 cm^−1^, and 1537 cm^−1^ correspond to the stretching vibration of N−H, C=O, and C−N, respectively. The asymmetrical and symmetrical stretching peaks of alkyl −CH_2_− moieties appear in 2950~2860 cm^−1^. The strong peak at 1249 cm^−1^ belongs to the C−O stretching vibration of urethane groups and ester moieties in polycarbonate diol [26,27]. These characteristic peaks suggest the formation of the urethane (−NHCOO−) structure. Moreover, the characteristic absorption peaks that were ascribed to the NCO and −OH groups were not found at 2270 cm^−1^ and 3500 cm^−1^, indicating that isocyanate and hydroxyl groups have completely converted into urethane bonds. Compared with CPU, ZPU appeared with new peaks at 1066 cm^−1^, which belonged to the stretching vibration of the S−O in sulfonic acid groups [28]. In addition, the weak absorption peak at 898 cm^−1^ was attributed to quaternary ammonium groups (N^+^−CH_3_) in the hydrophilic chain extender MDEA. These characteristic peaks prove the successful synthesis of zwitterionic polyurethane.

The surface chemistry of the ZPU film was further detected with an XPS test. As presented in Figure 3a, the signals of the C, O, N, and S elements were all detected from the ZPU film. N and S belonged to zwitterionic moieties (−N^+^ (CH_3_) − and −SO_3_^−^). The high-resolution spectrum of N 1 s (Figure 3b) was divided into two peaks at 399.8 eV and 402.2 eV, which were then assigned to nitrogen in urethane groups (−NH−COO−) and the positively charged amino groups (−N^+^) in zwitterion, respectively. The S 2p narrow spectrum (Figure 3c) containing spin−orbit doublet splits of S 2p_3/2_ (167.8 eV) and S 2p_1/2_ (168.9 eV) corresponded to the sulfonate groups (−SO_3_^−^). Moreover, the high-resolution C 1 s (Figure 3d) could be divided into C–C/C–H (284.7 eV), C–S (286.5 eV), and O=C–O (288.9 eV), respectively. These peaks were consistent with those in previous references [17,29]. All the above results further confirmed the presence of zwitterionic groups in ZPU.

### 3.2. Thermal Property

The thermal stability of polyurethanes is a significant parameter because their application processes may be conducted at high temperatures. The thermal stability of ZPU and CPU was compared by thermal weight loss analysis (TGA) and differential thermogravimetric analysis (DTG), as presented in Figure 4. The two polyurethanes show a similar thermal decomposition process. First, a slight weight loss occurred below 260 °C, which was probably ascribed to the evaporation of moisture and residual HAc or PST. In addition, the main decomposition stage of the two polyurethanes appears in the range of 260–450 °C, which is attributed to the cleavage of urethane and carbonate bonds. The maximum weight loss rate temperature of ZPU (335 °C) reduced slightly, compared with CPU (345 °C). A similar phenomenon was also reported in the polyurethanes containing sulfobetaine groups [24] . The small difference in decomposition temperature indicates that zwitterionic moieties had no significant effect on the stability of polyurethane.

### 3.3. Mechanical Properties

It is known that the mechanical properties of polymers are strongly influenced by physical or chemical cross-linking. The mechanical properties of ZPU and CPU were investigated by comparing their tensile strength, elongation, and Young’s modulus (Figure 5a and Table 1). It can be seen that the tensile strength and elongation of ZPU are 7.38 ± 0.45 MPa and 980 ± 21%, which are distinctly higher than that of CPU (1.32 ± 0.20 MPa and 760 ± 18%), respectively. It is known that there are abundant hydrogen bonds among urethane groups and microphase separation in conventional polyurethane, which endows the CPU with a degree of tensile strength and stretchability. As the zwitterionic groups are introduced, stronger physical cross-links among molecular chains of ZPU can be formed through the electrostatic interactions, thus further improving the tensile strength and stretchability of ZPU. It should be noticed that excellent stretchability usually represents the high chain mobility or flexibility of polymers which play an important role in self-healing and balancing the mechanical strength and healing efficiency. However, Young’s modulus of ZPU (6.75 MPa) is slightly lower than that of CPU (7.78 MPa). This phenomenon is mainly due to the fact that the linear CPU behaves with a crystallization ability, which is conducive to the stiffness of the polyurethane but tends to make the material brittle [30]. As the zwitterionic groups were incorporated, stronger physical cross-links were formed among polyurethane chains, and the crystallization was broken, making the polyurethane tough and flexible. Therefore, the CPU presents a higher Young’s modulus but a lower toughness than ZPU. Moreover, the elastic recovery property of ZPU and CPU was also compared, as shown in Figure 5b. The ZPU film stretched to 300% and could almost recover to its original length in 2 min after releasing the external force. Comparatively, the stretched CPU film still showed an elongation of about 98% even after 80 min of rest, showing evident inferior elastic restorability. This phenomenon indicates that the ZPU film shows better elastic restorability than CPU due to the synergistic effect of electrostatic interactions and hydrogen bonds.

In practical applications, polyurethane materials are usually required for the fast recovery of their initial shape and properties after a loading process. The elastic recovery property of ZPU is further evaluated by cyclic tensile tests with different strain ranges and time intervals. Firstly, the cycle tensile test of ZPU was conducted by sequentially increasing the maximum strains from 200% to 600% without waiting times between each cycle (Figure 6a). It can be seen that ZPU shows large hysteresis loops after successive cycles regardless of different maximum strains because of the broken dynamic ionic bonds, which need more time to be reorganized at new sites during stretching. The energy dissipation efficiency is defined as the ratio between the integrated area of the hysteresis loop and that of the loading curve [31]. It is calculated that the energy dissipation efficiency of ZPU under 200–600% maximum strains is 70%, 76%, and 72%, respectively. This dissipative feature from small to large deformations is enabled by the progressive sacrifice from weaker hydrogen bonds to stronger electrostatic interactions.

Further, ten consecutive loading-unloading tests were performed at a constant strain of 200% to further reveal the role of dynamic reversible bonds in dissipating energy (Figure 6b). A large hysteresis ring area appeared during the first loading and unloading process, which was due to the energy dissipation of ionic bonds. The energy dissipation in the second cycle was much smaller than that in the first cycle because the electrostatic interactions and hydrogen bonds after fracture did not have enough time to reorganize and restore to their original state. In addition, the hysteresis ring area appeared slightly reduced in the subsequent cycles, meaning that electrostatic interactions and hydrogen bonds underwent rapid reconstruction to keep the initial structure of ZPU. Moreover, with a sufficient relaxation time (30 min), the stress–strain curve almost overlaps with the first cycle, and the residual strain, stress, and hysteresis energy were also recovered. The above results demonstrate that ZPU has an outstanding capability to recover its initial shape after deformation. The dynamic feature was conducive to extending the service life of the polyurethane material in practical applications.

### 3.4. Self-Healing Property

Apart from dissipating the stress energy, the non-covalent ionic interactions with low binding energy also worked as dynamic bonds, providing the effective reconstruction of materials upon damage. The self-healing property of the CPU and ZPU film was quantitatively evaluated via the tensile test of the original sample and the healed samples from a fully cut state. Here, we mainly discussed the influence of the healing time on the recovery of mechanical properties at 50 °C, and the healing process was observed with a digital microscope. The CPU film, after healing for 1.5 h and even 48 h was too weak to conduct a tensile test. Figure 7a shows the stress–strain curve of the ZPU film self-healed at 50 °C for a different time. It can be calculated that the self-healing efficiency was only 40% and 58% at 50 °C for 1 h, while the film recovered about 93% and 97% of its original tensile strength and breaking elongation after 1.5 h, respectively. This result illustrates that extending the healing duration is conducive to improving healing efficiency. From the optical microscopy images of the cut ZPU film before and during the healing process, it could be observed that the crack almost disappeared after healing at 50 °C for 1.5 h (Figure 7b). Compared with the control of the CPU film, the ZPU film could effectively recover to its original mechanical property under the same condition. In view of the other same structure of CPU and ZPU except for the different ion types, it could be inferred that the ionic interaction from zwitterion in ZPU endowed the self-healing function. The self-healing mechanism is as follows: when two water-moistened pieces of ZPU film are subjected to heat treatment, the polyurethane chains are softened and move quickly to the scratch surface via the involvement of water and heat; meanwhile, the broken ionic bonds among the protonated ammonium moieties and sulfonic groups in the polyurethane chains recombine at new sites via electrostatic interactions, thus repairing the damage.

### 3.5. Reprocessing Ability

Due to the dynamic nature of zwitterionic bonds, ZPU can be reprocessed by conventional solution casting and hot-pressing methods. For recyclability by solution casting, the ZPU film was cut into small pieces, and then dissolved in THF at room temperature for 12 h, followed by pouring into a Teflon mold. The zwitterionic polyurethane network was re-formed after drying at room temperature for 48 h, and then at 60 °C for another 24 h (Figure 8a). As for reprocessing by hot-pressing, small fragments of the film were heated and compressed at 80 °C, 8 MPa for 5 min (Figure 8b). It can be seen that the regeneration of homogeneous films from both the solution casting and hot-pressing methods show a similar appearance and transparency with original films. The tensile properties of the recycled films were also measured to evaluate the recovery efficiency, as presented in Figure 8c. It can be observed that the strain-stress curves of the recycled and original samples were similar, although the tensile strength and breaking elongation decreased slightly which may be caused by small amounts of solvents and small pores in the reshaped ZPU film. Similar phenomenon can also be found in other references [30]. The recovery efficiency for tensile strength and the breaking elongation of the films recycled by solution casting are 89% and 100%, respectively, while the recovery efficiency of the films reprocessed by hot-pressing are 98% and 88%. The results illustrate that the recombination of dynamic electrostatic interactions effectively endows the ZPU film with excellent recyclable capability. Similar to the self-healing process, the zwitterionic bonds on the surfaces of the tiny fragments regenerate new electrostatic interactions to promote the reconstruction of the PU networks, thereby achieving recyclability and reshaping.

## 4. Conclusions

In summary, a zwitterionic polyurethane (ZPU) with outstanding self-repairing and recyclable capabilities was successfully synthesized. Zwitterionic moieties have no significant effect on the stability of polyurethane. Moreover, the zwitterionic groups can form strong physical cross-links among molecular chains via the electrostatic interactions while in the meantime, dissipating strain energy as a weak dynamic bond, which endows the polyurethane with excellent mechanical properties. As a result, ZPU exhibits a robust tensile strength of 7.38 MPa, with an outstanding stretchability of 980%, Young’s modulus of 6.75 MPa, and excellent elastic recovery properties. The tensile strength and elongation at the break of a fully cut ZPU sample recovered over 90% of the original values at 50 °C for 1.5 h due to the dynamic reconstruction of reversible ionic bonds. Meanwhile, the recovery efficiency was more than 88% for the solution casting and hot-pressing (80 °C, 8 MPa, 5 min). This work provides a good reference for developing polyurethane materials with outstanding mechanical, self-healing, and recycling properties through the widely used resource and simple synthesis process. In addition, the ZPU is a promising candidate for application in durable protective coatings, wearable electronic devices, and other sustainable industrial applications.

## Figures and Tables

**Figure 1 polymers-15-01270-f001:**
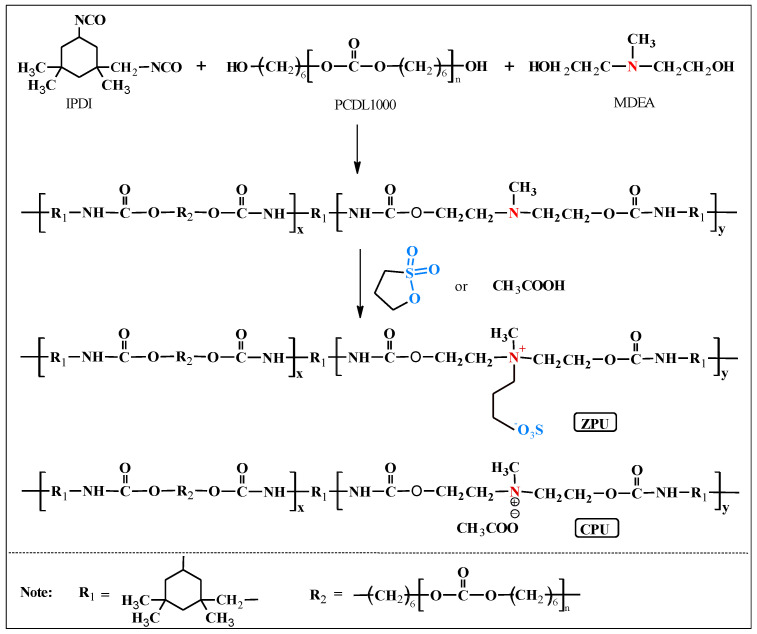
Synthetic routes of ZPU and CPU.

**Figure 2 polymers-15-01270-f002:**
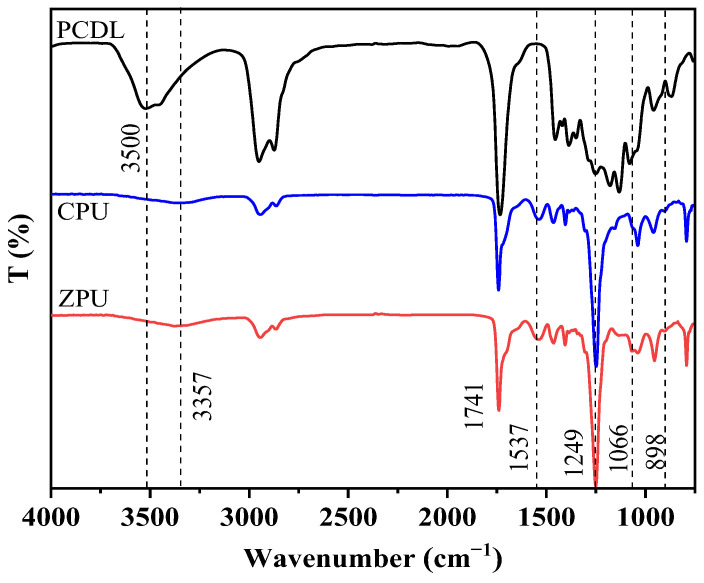
FTIR of ZPU and CPU.

**Figure 3 polymers-15-01270-f003:**
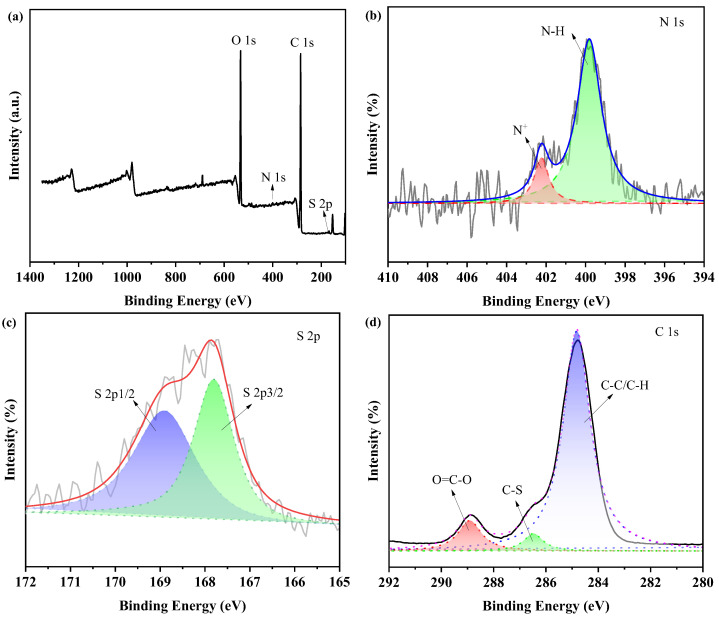
(**a**) XPS wide spectrum of ZPU; high-resolution XPS spectra of the N 1s (**b**), S 2p (**c**), and C 1s (**d**), respectively.

**Figure 4 polymers-15-01270-f004:**
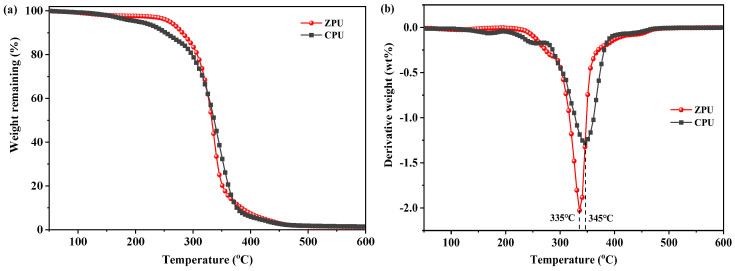
Thermal property of ZPU and CPU: (**a**) TGA; (**b**) DTG.

**Figure 5 polymers-15-01270-f005:**
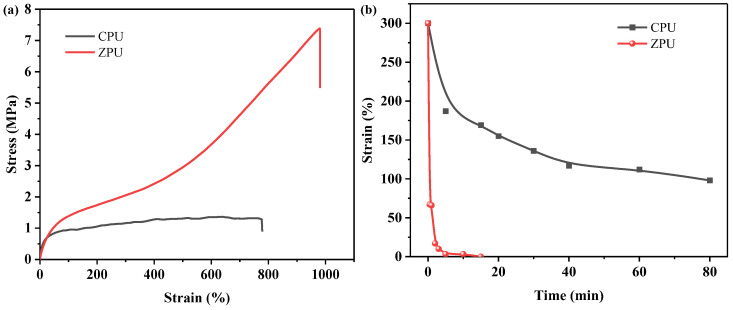
Stress–strain curve (**a**) and elastic recovery property (**b**) of ZPU and CPU.

**Figure 6 polymers-15-01270-f006:**
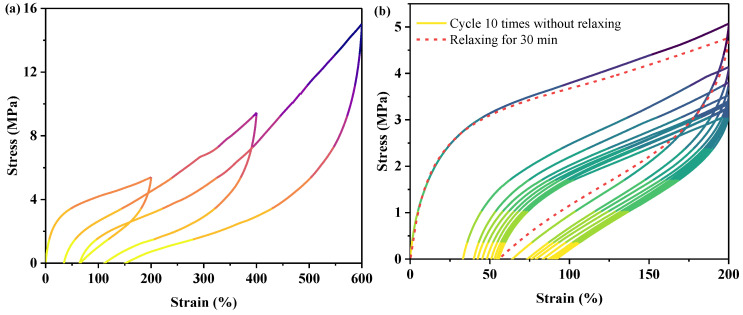
Continuous cyclic tensile curves of ZPU (**a**) At various strains without relaxing, (**b**) At 200% strain without relaxing and then relaxing for 30 min.

**Figure 7 polymers-15-01270-f007:**
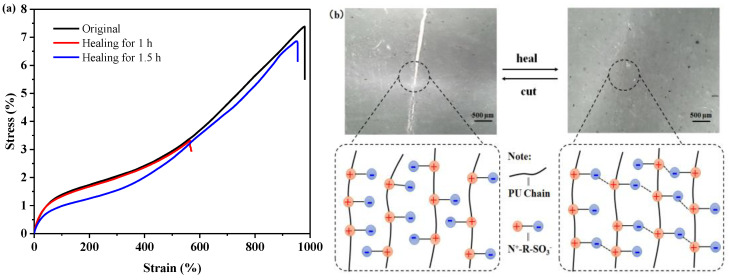
(**a**) Stress–strain curves of the original and healed ZPU films; (**b**) Optical microscopy images of scratched and healed ZPU films and self-healing mechanism.

**Figure 8 polymers-15-01270-f008:**
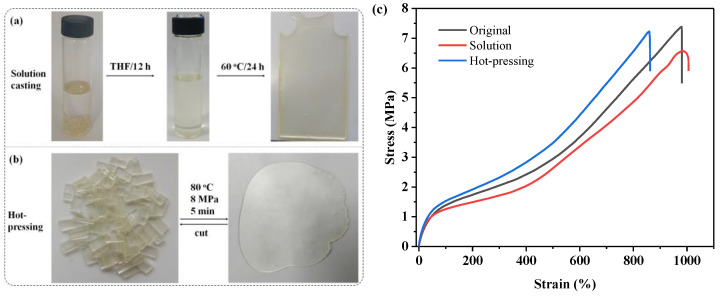
Recyclability of ZPU film: (**a**) Photographs of ZPU soaked in THF without heating; (**b**) Photographs of ZPU reprocessed by hot-pressing; (**c**) Strain–stress curves of the original and recycled ZPU samples.

**Table 1 polymers-15-01270-t001:** Comparison of tensile strength, elongation at break and Young’s modulus.

Samples	Tensile Strength (MPa)	Elongation at Break (%)	Young’s Modulus (MPa)
CPU	1.32 ± 0.20	760 ± 18	7.78 ± 1.01
ZPU	7.38 ± 0.45	980 ± 21	6.75 ± 0.11

## Data Availability

The data presented in this study are available on request from the corresponding author.

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
