# Peer review of "A Self-Healable and Recyclable Zwitterionic Polyurethane Based on Dynamic Ionic Interactions"

_polymers, 2023, doi:10.3390/polym15051270_

Round 1

Reviewer 1 Report

The authors developed a self-healing and recyclable zwitterionic polyurethane (ZPU) by introducing ionic bonds between protonated ammonium groups and sulfonic acid groups. There are some issues that need to be clarified by the authors.

1. Only the synthesis and structure of ZPU are given in Figure 1, but the full name and structure of WPU are not clear.

2. In line 116, in infrared analysis 1, please confirm whether the wave number 1249 is the stretching vibration peak of the carbonyl group.

3. The tensile test in Figure 5c should show the point of inflection where the sample breaks.

4. Corresponding to the tensile test in Figure 5, how many times were tested for each sample? The results should be tabulated, indicating error, and calculate Young's modulus.

5. Figure 7b lacks the ruler for the light microscope picture.

6. There are grammatical problems in part of the writing and need to be revised.

Reviewer 2 Report

The authors study a self-healing polyurethane including zwitterion, N+ cations and SO3- anions on the backbone. The subject is suitable for Polymers, but there are uncertain descriptions in this manuscript. I judged this manuscript is major revision. My comments are as follows.

 1. About an abbreviation of PS. It is better to use another abbreviation because one generally recognizes PS as polystyrene.

 2. Page 5, line 150 to 152: Authors describe “the 50% thermal weight loss temperature (T50%) and the maximum weight loss rate temperature of ZPU are reduced by 3 °C and 10°C,” In general, a differential peak will be given at the large slope of TGA. Figure 4a shows both slopes at T50% looks like the steepest line. It is requisite to explain why such large difference is produced, and show the values of weight loss at the peaks of DTG.

Furthermore, it is necessary to explain the sentence about “ZPU shows higher thermal stability below 300°C, and slightly low thermal stability above 300°C” in detail. What causes such results? Does the difference of only 3°C suggest the stability below 300°C? Does the gradual weight loss of WPU rather than that of ZPU over 300°C support the stability above 300°C? What contributes the thermal stability.

 3. Page 5, line 156 to 171 in page6: I require the color of ZPU in Figure 4 to be red because ZPU should be conspicuous rather than WPU, and moreover the colors between Figures 4 and 5 should be the same. In addition, the description from line 156 to 171 is completely wrong. Description and figures are not consistent. The authors use ZPU and WPU like a random. Readers do not understand which PU is superior for recovery and show self-healable properties.

 4. Do the authors find any evidence of the ionic interaction? If not, the sketch in Figure 7 is a hypothesis. It is necessary to show the evidence of the ionic interaction. There is no description about the interaction on FTIR spectrum.

 5. Figure 8c: Why does the film obtained from the THF solution not show the original SS curve? The SS property of the film obtained from hot-pressing shows lesser elongation and strength values than those of original one. I think this property is reasonable because the scratched region will be weaker than that of the original in the case of hot-pressing; no complete ionic interaction is formed. If the healing is achieved by the ionic interaction only, the film from the solvent casting will show the same SS property as the original, I guess. Is the reduction of 11% for tensile strength within an experimental error?

Round 2

Reviewer 1 Report

The authors have addressed all my questions and the manuscript can be accepted.

Reviewer 2 Report

Authors revised the manuscript comprehensively  in response to my inquiries.

I agree this manuscript to publish in "Polymers".